# Biomolecules involved in the metabolism of *Escherichia coli* affected by photodynamics: A calorimetry study

Daniel Ortega-Zambrano[1], Citlalli Lona-Yepez[1], Francisco J. Sierra-Valdez[2]*, Hilda Mercado-Uribe [1]*

**1** Centro de Investigación y de Estudios Avanzados del Instituto Politécnico Nacional Unidad Monterrey, Vía del Conocimiento 201, Parque PIIT, Apodaca, 66600, Nuevo León, México, **2** Tecnológico de Monterrey, Escuela de Ingeniería y Ciencias, Monterrey, Nuevo León, México

\* hmercado@cinvestav.mx (HMU); f.sierra@tec.mx (FJSV)

## Abstract

Antimicrobial resistance (AR) is a global health problem with significant consequences for the population and the economies of governments. In this context, several efforts are being made to investigate and develop alternative methods to counteract this situation, for example, photodynamic inactivation (PDI), which is a non-specific treatment to inhibit pathogenic microorganisms. It is based on the excitation of a photosensitizer molecule (PS) with UV-Vis radiation to generate reactive oxygen species (ROS), employing molecular oxygen already available in the environment. Due to their high reactivity, ROS produce oxidation of lipid membranes, proteins, and nucleic acids, eventually leading to cell death. Despite the fact that multiple works have been carried out using PDI, the investigation about the structural changes induced in the biomolecules of microorganisms that lead to cell inactivation has been limited. In the present work, we used differential scanning calorimetry (DSC) to study the thermodynamic changes produced by PDI in *E. coli*. We showed that such changes are correlated with the loss of viability and metabolic processes.

## Introduction

Antimicrobial resistance (AR) is a natural process that occurs when pathogenic microorganisms are exposed to toxic drugs [1,2]. The concerning global escalation in AR requires innovative strategies to confront this situation. In general, the problem has been addressed by focusing on three different perspectives: variations in existing antibiotics, the development of new drugs, and the work on complementary therapies [3–5]. In this context, photodynamic inactivation (PDI) has been a promising approach that takes advantage of the entire spectrum of light to eradicate a wide variety of pathogens [6]. PDI is based on the use of three components: a photosensitizer molecule (PS), light of an appropriate wavelength, and molecular oxygen in

**Data availability statement:** All relevant data are within the manuscript and the Supporting Information files.

**Funding:** This work was funded by CONACyT, Mexico, Grant number A1-S-8125 (HMU). DOZ and CLY were supported by fellowships by CONACyT, Mexico. https://secihti.mx/ The funders had no role in study design, data collection and analysis, decision to publish, or preparation of the manuscript.

**Competing interests:** The authors have declared that no competing interests exist.

the medium. The joint action of these elements generates several reactive oxygen species that inactivate unwanted pathogens [7]. Despite the growing success of this method, there is limited exploration of the structural changes induced in the biomolecules of microorganisms that lead to cell inactivation.

Differential scanning calorimetry (DSC) is a non-perturbing thermodynamic technique, mainly used to determine differences in enthalpy due to conformational changes of individual biomolecules, obtained with a thermogram (thermal fingerprint) [8]. DSC also provides a robust understanding of biomolecular interactions and their alterations through physicochemical perturbations. In complex systems, such as microorganisms, DSC has been used to explore the thermotropic behavior of bacteria using their unique multipeaked spectrum as a result of the structural states of all biomolecular entities comprised. This thermal-structural relationship is obtained from the overall heat-induced cooperative transitions of lipid membranes, proteins, nucleic acids (DNA and RNA), and polysaccharides in balance with the environment. In this regard, any structural change that compromises the folding of a particular biomolecule and/or the intermolecular interactions among them can be identified. For instance, Mackey et al. [9] have reported the thermal damage that led to the death of *E. coli* NCTC8164, separating the main components of the thermogram to identify the nature of the calorimetric peaks. The result showed an irreversible denaturation process identified by some specific ribosomal subunits (30S-70S), and the presence of endothermic peaks associated with tRNA and DNA. Analogously, a cold shock procedure was induced in *Listeria monocytogenes* to investigate how the thermal stability is disturbed. *L. monocytogenes* is a gram-positive bacterium characterized by growing in reduced-water environments, as well as in a wide range of temperatures (1–45 °C). The tolerance to cold-stress was reduced mainly due to the melting transitions of ribosomes [10]. Other alternatives to a thermal process have been proposed to eradicate pathogens. For example, the application of high hydrostatic pressure was also evaluated by DSC in diverse bacteria. A comparison between *E. coli* and *S. aureus*, whose sensitivities to this type of stress is different, was performed by Alpas et al. [11], and independently by Kalentuc et al. in *Leuconostoc mesenteroides* [12]. Ribosomal denaturation was the fundamental response that led to cell death and changes in the DNA melting transition as a function of pressure [11,12]. In addition, structural modifications (decreasing of the bacteria chain and blister formation on the cell surface) were observed in the case of *L. mesenteroides*. Other authors have used different chemical agents (ethanol, salts and very acidic pHs) to characterize the thermodynamical transitions of the main components of *E. coli* [13]. This procedure (known as hurdle technology) showed a reduction in the thermal resistance of the pathogens. In another study [14], *A. oxydans*, a soil bacterium that changes its morphology depending on the growth stage, was exposed to potassium permanganate (PM). It produces the $MnO_4^{-2}$ ion, which is a strong oxidant and is used for soil remediation. The analysis of the whole cell by DSC showed that the membrane cell and DNA were disturbed and that a possible ribosomal denaturation was exhibited. More recently, Brannan et al. [15] used the antimicrobial peptide (AMP) MSI-78 to investigate its interaction with non-lipidic targets in *E. coli*. AMP as an essential component of the immune system and its effectiveness against multi-drug resistant

pathogens. Measurements by DSC probed that MSI-78 has two main mechanisms of action in bacteria, it constrains the ribosomal activity and disturbs their cell membrane.

In general, it is clear from the previously commented studies that calorimetry approaches are useful when exploring biomolecular changes in bacteria under diverse treatments.

In the present work, we used DSC to investigate the thermal structural changes caused by photodynamic inactivation (PDI) in *E. coli* K12-MG 1655 [5]. We showed that such disturbances are correlated with loss of viability and exothermic regions due to metabolic processes. We also found interesting PDI effects by irradiation alone (without photosensitizers), emphasizing the intrinsic impact of specific wavelengths on some biomolecular structures.

## Materials and methods

### *E. coli* culture preparation

The pathogen studied in this work was the *E. coli* K-12-MG-1655 strain. It was previously stored in a Luria Bertani (LB) broth at −2 °C. First, a sample of this saturated bacteria suspension was poured into 20 mL of Luria Bertani (LB) medium and incubated at 37 °C in an orbital shaker for 16 h at 130 RPM. The suspension was incubated at 37 °C for 6 h at 130 RPM, reaching the exponential phase known for *E. coli*. This means that the optical density (OD) of the suspension was 0.3 (equivalent to $1.5 \times 10^9$–$1.8 \times 10^9$ colony-forming units (CFU)/mL), which was measured with a spectrophotometer (Multiskan GO, Thermo Scientific). Next, 45 mL of the bacterial culture was poured into an eppendorf tube and centrifuged for 10 min at 4000 RPM. The supernatant was then discarded, and a pellet was obtained. The procedure was repeated five times until the concentrated biomass was obtained. This suspension was centrifuged at 4000 RPM for 10 min, washed, and re-suspended in 10 mM phosphate buffer saline (PBS) and again centrifuged three times. The optical density corresponded to $3.6 \times 10^{11}$ CFU/mL.

### Control and experimental samples

To prepare the experimental samples, 1 mL of concentrated bacteria was poured into a 24-well microplate. To prevent evaporation, the empty spaces in the microplate were filled with milli-Q water. The microplate was then covered and sealed with parafilm. The temperature during the irradiation process was monitored using a thermocouple. At the end of the process, it increased 4.4 °C. Then, the thermomagnetic stirrer setpoint was adjusted to 32.6 °C (4.4 °C below the culture temperature) to compensate for the increase in temperature due to radiation.

All samples were prepared and measured in triplicate. As an initial control, we considered incubated bacteria (IB) kept in dark conditions and stirred at 200 RPM and 37 °C for 24 hours. Another group of samples was analyzed using DSC immediately after re-suspension in PBS. We refer to them as non-incubated bacteria (NIB).

The effect of 47 mM methylene blue (MB, 102520851, Sigma-Aldrich) was also explored in cell cultures. We exposed a group of samples to red light (660 nm), which we called (IB + RL + MB). It is known that many types of bacteria, including *E. coli*, contain endogenous sensitizers (porphyrins and flavins), which upon exposure to UVA and blue light (405 and 450 nm) produce ROS. They induce membrane and DNA damage, as well as lipid peroxidation [16–18]. As a control experiment, we irradiated a group of bacteria with UVA radiation to ensure that the obtained results were not due to the response of endogenous sensitizers. We refer to this sample as IB + UVA. Both types of samples, IB + RL + MB and IB + UVA, were independently irradiated for 24 h at 37 °C. The following combinations are also included: non-incubated samples (NIB) and incubated samples (IB) with the addition of MB (NIB + MB and IB + MB), as well as incubated samples without MB and irradiated with RL (IB + RL).

### Photodynamic treatment

Irradiation was performed on a 24-well microculture plate using two laser diode modules (Laserland) of 405 nm and 660 nm wavelengths, with a nominal optical power of 150 mW. The laser spot size and FWHM (Full Width at Half Maximum)

for the 405 and 643 nm laser were 4.88 mm and 1.27 nm, and 5.96 mm and 13.3 nm, respectively. The inner diameter of each well was approximately 15.5 mm. The irradiance was 80 mW/cm$^2$ and the total fluence 6912 J/cm$^2$. The distance between the light source and well was 30 cm. To ensure homogeneous exposure of the sample (1 mL per well), it was maintained under continuous agitation at 200 RPM with a magnetic stir bar.

### Colony-forming units count

The count of the colony-forming units (CFU) was performed by 7–8 serial dilutions in a PBS buffer (a dilution factor of 10). For each sample, 100 $\mu$L of the bacteria suspension was poured into a Petri dish with 20 mL of liquid LB agar. This suspension was homogenized by shaking and incubated for 48 h at 37 °C. Each solution was plated in triplicate and the CFU per milliliter were counted.

### Differential scanning calorimetry

The calorimetry profiles or thermograms were obtained using a NanoDSC microcalorimeter (TA Instruments) that measures the heat capacity ($C_p$) at a constant pressure (3 atm) as a function of temperature with a scan rate of 1 °C/min. Previously, all samples were degassed under mild vacuum (635 mmHg) for 15 min at room temperature. After a few minutes of equilibration at 5 °C, thermograms were acquired. $C_p$s values were obtained after subtracting the baseline signals. They correspond to spin lines from a phosphate buffer saline solution (PBS). To obtain more defined lipid and protein peaks, we performed a first run from 2 °C to 95 °C, followed by two runs from 2 to 45 °C. The aim of the first run was to fully denature proteins that are in contact with lipids. The second and third runs were performed to investigate the transition temperature of the lipid membrane. To find more subtle changes induced by the photodynamic process in the calorimetry profiles, three temperature ranges were investigated separately: 10–40 °C, 30–70 °C and 70–85 °C. Two parameters, the change in enthalpy ($\Delta$H) and the temperature where $C_p$ is maximum ($T_m$), were obtained from the calorimetry profiles. Each type of measurement was performed in triplicate to ensure reproducibility. The final values came from the average of these measurements. Data were analyzed using the software provided with the calorimeter and MATLAB. $\Delta$H, the area under the curve, was calculated by integrating the molar heat capacity of the initial and final temperature ($T_i$ and $T_f$, respectively) in each region of the thermogram.

### Statistical analysis

The Shapiro-Wilk test showed that the data do not follow a normal distribution. In addition, the Wilcoxon signed-rank test was applied to evaluate statistically significant differences between the experimental and control groups. The significance levels were established at p ≤ 0.1 for *.

### Results and discussion

As mentioned in the Introduction, studies of *E. coli* cells have been carried out using differential scanning calorimetry (DSC) for different purposes, from the acquisition of kinetic parameters during the inactivation of microorganisms [19], to the analysis of the thermal stability and reversibility of certain biomolecules [20]. The aim of the present work was to evaluate the thermodynamic response of *E. coli* bacteria (K12-MG1655) subjected to photodynamics. Fig 1 shows the representative calorimetry profiles after the baseline is subtracted for the experimental sample IB + RL + MB, at constant pressure. The profiles are compared with the control case consisting of bacteria incubated in PBS for 24 hours under dark conditions at 37 °C (IB). It is well documented that bacteria lipid membrane transitions occur below their incubation temperature [21]. In fact, our results showed that in this first run, the peak of lipids is not evident in the control sample (see the yellow thermogram in Fig 1 below 40 °C). However, in the other two groups, IB + UVA and IB + RL + MB, the lipid peaks clearly appear around 33 °C. It seems that the diversity of ROS generated by irradiation induced irreversible damage to

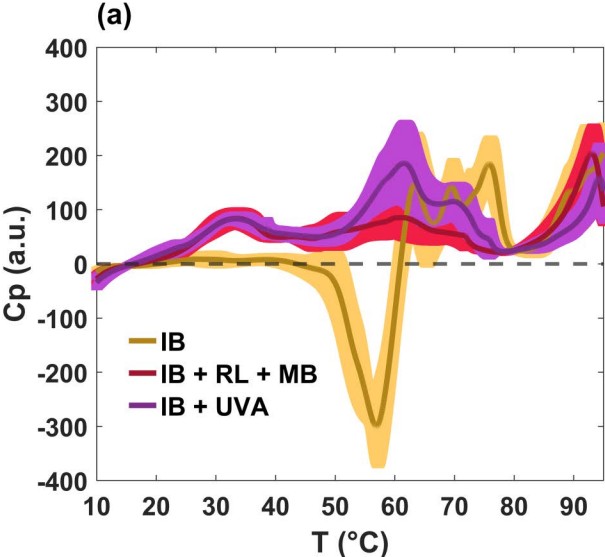

**Fig 1. Differential scanning calorimetry profiles of *E.coli* bacteria (control and experimental): heating scan.** The control sample corresponds to an incubated culture (IB, yellow line). IB+UVA (magenta) and IB+RL+MB (red) are the experimental samples, which were illuminated with UVA (405 nm) and red light (600 nm), respectively, for 24 hours. The emergence of the lipid peak after irradiation is evident, and the most striking effect in the control sample is the existence of an exothermic valley. Three repetitions of each measurement were performed. The averages are displayed by the lines, while the shadow bands correspond to standard deviations.

crucial structures, such as proteins and lipids. In particular, unsaturated acyl chains are more susceptible to oxidation [22,23]. A significant change is also observed in the region between 40 and 80 °C, which can be attributed to cell wall and/ or protein damage. Regardless of the precise oxidative effects, the photoinactivation process appears to increase the cooperativity of the lipid melting. Furthermore, the three cases showed a similar high-temperature profile associated with DNA unfolding at temperatures greater than 90 °C [20].

A characteristic worth highlighting is the existence of an exothermic valley around 58 °C in the control sample (see the yellow line in Fig 1, which is completely removed by the photodynamic process. This effect is indicative of a metabolic process [24–26]. It is well-known that all the reactions that occur within cells during metabolism produce heat. DSC allows us to measure the heat flux inside and outside the cell in a temperature-controlled environment. Energy absorption processes (endothermic) are exhibited by peaks in the calorimetry profile; for example, protein denaturation. On the other hand, energy releasing processes (exothermic) are exhibited by valleys due to the dissipation of heat that is generated by the intracellular dynamics; for example, enzyme kinetics, cell replication, and metabolism. Fig 2a shows an amplification of this exothermic valley, as well as the corresponding ΔH (area under the curve calculated from 45 to 60 °C) which is equivalent to the calorific capacity and viability of the sample obtained by the count of CFU (see Fig 2b). It can be seen that positive values in ΔH are associated with no viability in the experimental groups, while negative values correlate with it. It is clear that the IB+UVA and IB+RL+MB profiles do not have a valley, implying the death of the pathogens.

These results persuaded us to explore the effect of the aging time (24 hours) of the control sample. We will return to this point later.

To investigate in detail the structural changes of bacterial biomolecules caused by PDI, we analyzed different temperature ranges of the thermogram. Fig 3 shows the region of protein unfolding, as well as the corresponding changes in enthalpy for the previous samples (IB, IB+UVA and IB+RL+MB). It is evident that there is a decrease in the height of the characteristic peaks after irradiation and a significant difference in enthalpy with respect to the control (IB). The negative

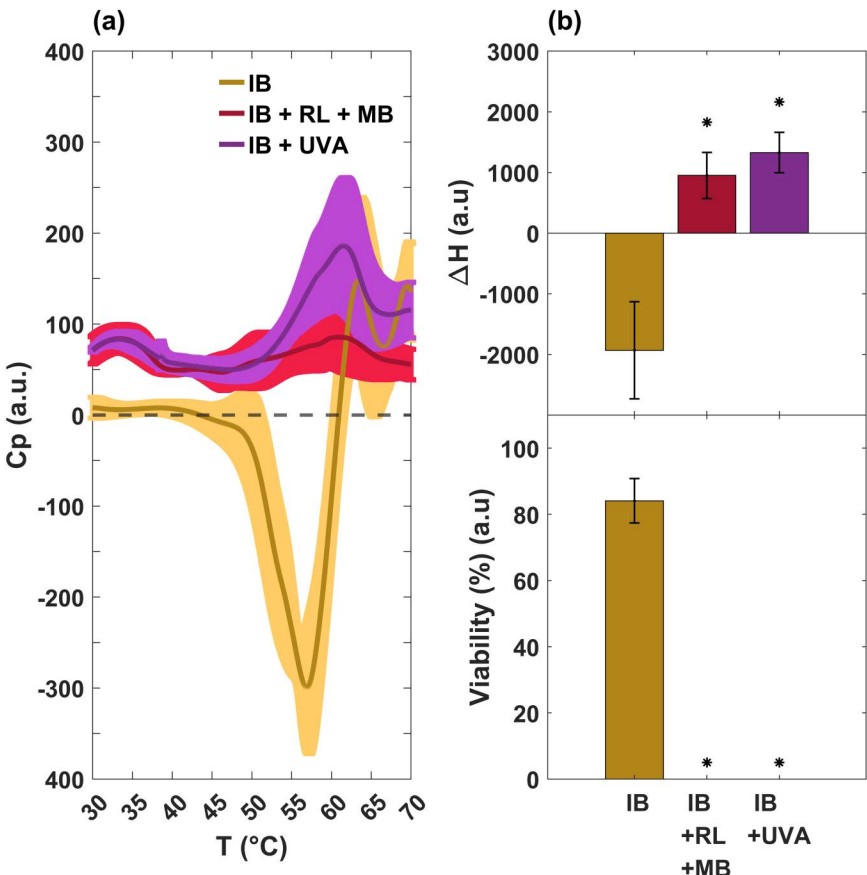

**Fig 2. Cellular energetics and viability. (a)** Amplification of the exothermic valley in the DSC thermogram, see Fig 1. **(b)** Changes in the enthalpy (area under the curve) of the exothermic region (top panel) compared to the viability of cell culture after the respective treatment (bottom panel). Positive values in enthalpy difference (absorption of heat) determine the absence of bacteria viability, while negative values in enthalpy difference (release of heat) imply bacteria viability. Values are averages (thin lines) and standard deviations (shadow bands) of three measurements. *$p \leq 0.1$ by the Shapiro-Wilk test and compared to IB.

values in the enthalpy suggest that ROS generated by PDI can induce structural destabilization [27,28] in such a way that the energy required to denature the irradiated proteins is lower than in the control samples.

To obtain the transition temperature in the lipid response after the photoinactivation process, these signals were normalized after the third consecutive run (see Fig 4). A small shift to the right (around 2.5 °C, see vertical dashed lines) can be observed in the response of the sample IB + RL + MB, which is associated with greater rigidity in the membrane [29–32], possibly caused by the action of ROS that oxidized the lipids in the membranes of bacteria.

Subsequently, we investigated the effect of aging time in incubated cultures (Fig 5). Here, we add the calorimetry profiles of the control samples, including the case of the non-incubated cultures (NIB), which refer to the bacteria without any treatment or incubation; in other words, the bacteria that were immediately measured on the calorimeter.

We found that such cultures exhibit an almost flat region (from 2 °C to 45 °C) in the calorimetry spectra, where a small peak around 31 °C is occasionally observed, which corresponds to the lipid membrane of the bacteria.

No significant differences were observed in the protein and lipid profiles shown in Fig 6, except for the case of IB + RL. Furthermore, there are no changes in the transition temperatures, indicating that there is no damage to the lipids, and the protein enthalpies are basically the same in all cases. These results are consistent with those obtained

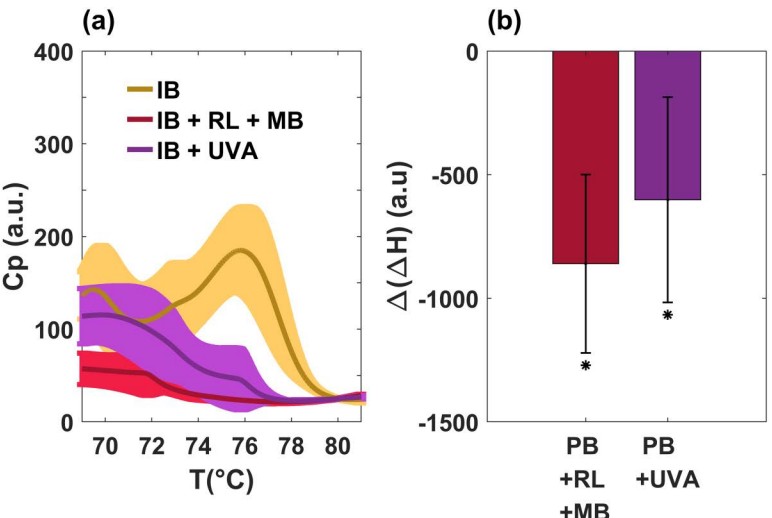

**Fig 3. Protein profiles and enthalpy change differences of *E.coli* bacteria obtained by differential scanning calorimetry during the first heating scan. (a)** There is a clear decrement in the height of the characteristic peaks after irradiation of the experimental samples IB+UVA (magenta) and IB+RL+MB (red) compared to the incubated culture (yellow line). **b)** There is a big difference in enthalpy respect to the control sample. Values are averages (thin lines) and standard deviations (shadow bands) of three measurements. *$p \leq 0.1$ by the Shapiro-Wilk test and compared to IB.

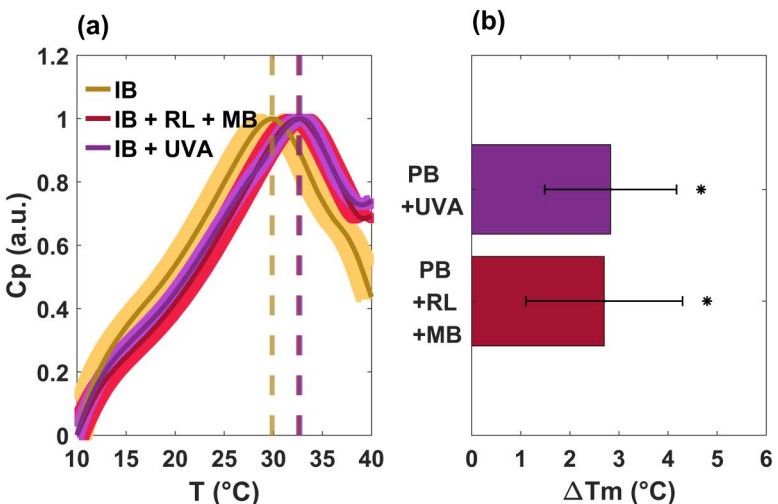

**Fig 4. Lipid profiles and transition temperatures of *E.coli* bacteria obtained by differential scanning calorimetry after the third heating scan. (a)** Calorimetry profiles of the lipid membrane region for cultures exposed to 405 nm (IB+UVA) and red light (600 nm) with methylene blue (IB+RL+MB), for 24 hours, and their comparison with the incubated control (IB). The average is displayed in a thin line; while the shadow band corresponds to the standard deviation. **(b)** $\Delta T_m$ corresponds to the change in the melting temperature (2.5 °C) of the lipids of the sample IB+RL+MB. Positive values mean a stiffening of the lipid membrane. The experiments were performed by triplicate. *$p \leq 0.1$ by the Shapiro-Wilk test and compared to IB.

in cultures in which viability decreases (as we will see later), although it can be attributed to lysis processes; from a thermodynamic perspective, proteins and lipids appear to remain intact. It is worth mentioning that there is inherent variability due to the biological nature of the samples. Variability is due to all the different treatments the samples undergo before entering the DSC. However, despite this intrinsic variability, significant differences can be observed in

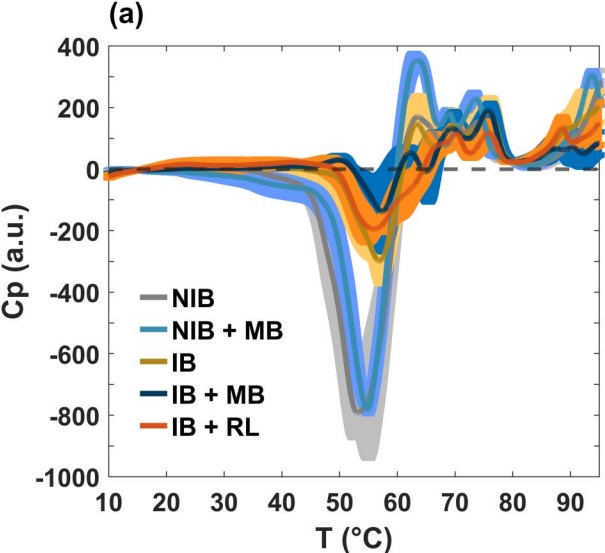

**Fig 5. Effect of aging time for incubated cultures.** Control samples correspond to both non-incubated (NIB) and incubated bacteria (IB), while experimental samples were treated with MB and red light (600 nm), for 24 hours. IB signal exhibits an exothermic peak smaller than that of NIB suggesting that these bacteria have lower metabolic activity, possibly due to bacterial aging and lysis of some of the cells. Three repetitions of each measurement were performed. The average is displayed in a thin line; while the shadow band corresponds to the standard deviation.

the sample we are truly interested, that is, in the irradiated culture. Additionally, it is important to note that the scale is greatly amplified.

We have previously mentioned that the interval between 45 and 59 °C shows a deep valley, which is only present in the control (non-incubated and incubated) bacteria. This is an exothermic process that is generally associated with cell metabolism. Although the incubated bacteria were suspended in PBS for 24 hours at 37 °C, and this reagent maintains the suspension at the regulated pH (7.0), it does not provide the necessary nutrients for their growth and reproduction. In other words, incubated bacteria are less metabolically active compared to non-incubated (NIB) cultures. For this reason, we call this calorimetry valley a metabolic valley.

To better appreciate the differences between the calorimetry profiles, a close-up of this region is shown in Fig 7a. We also present the correlation between enthalpy (area under the curve) and bacterial survival (see Fig 7b). Enthalpy values were obtained by integrating calorimetry profiles between 45 and 59 °C. Cultures with the highest viability are observed to have the most negative enthalpy values; meanwhile, less negative enthalpy values are associated with the lowest viability. Note that only the enthalpy of non-incubated cultures (gray lines) and non-incubated cultures treated with methylene blue (cyan lines) show a statistically significant difference with respect to incubated bacteria (copper lines). This result is consistent because both cases exhibit a deeper metabolic valley. Moreover, only the group of bacteria incubated with photosensitizer (IB+MB) exhibits a significant decrease in its population. This means that the photosensitizer produces a cytotoxic effect after the incubation period.

With respect to the exothermic valley, it is timely to say that, during the processes of cell reproduction and growth, the cell is extremely active. It is well known that bacterial cell division is coordinated by FTsZ, a tubulin homologue that is part of the cytoskeleton [33]. In cytokinesis, the FTsZ protein is combined with other proteins to form a ring-like structure, called the Z ring, which acts as a scaffold for the protein complex (divisome) responsible for cell division and manages the synthesis, location, and size of the septum. Subsequently, septation occurs through a notoriously cooperative process with more than 10 proteins working. The Z ring is highly dynamic; it constantly self-regulates and rearranges exchanging

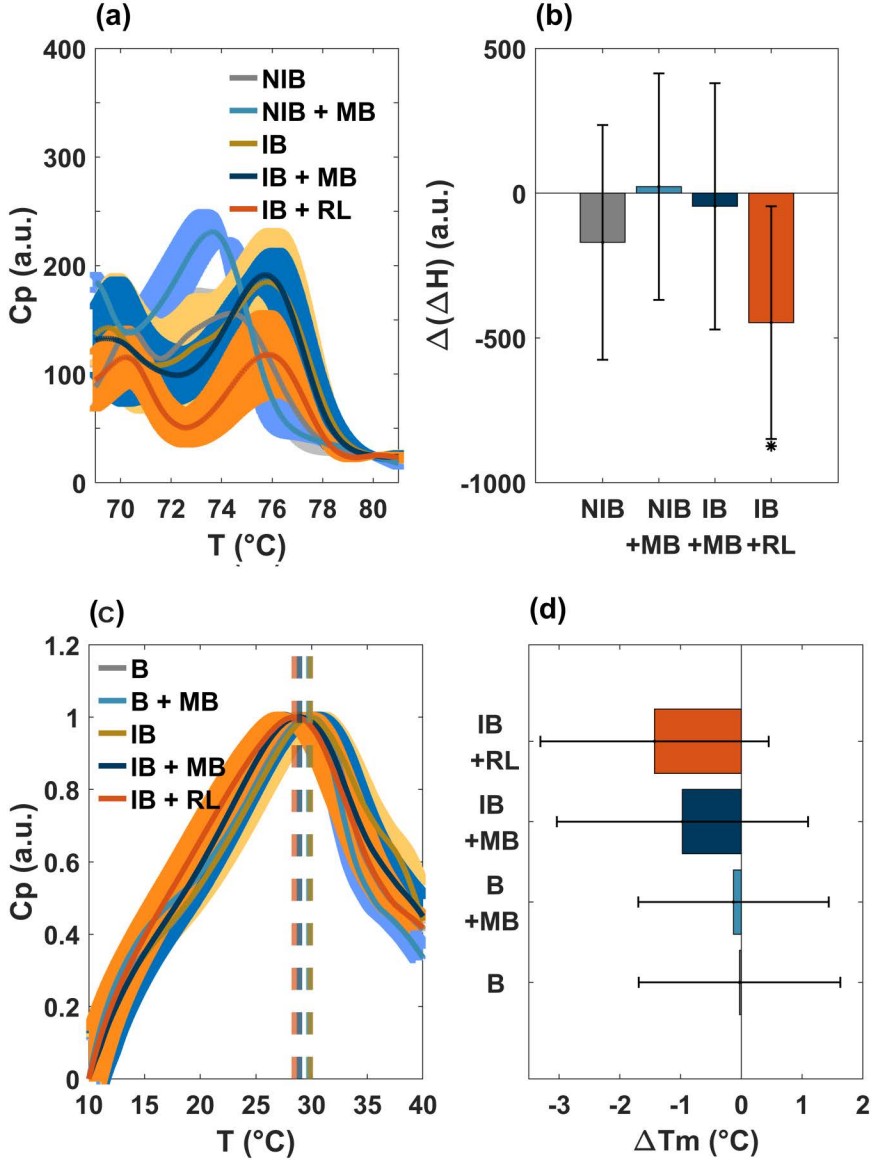

**Fig 6. Calorimetry results of non-incubated (NIB) and incubated *E.coli* culture (IB). (a-b)** Protein profiles and their corresponding enthalpy change differences. **(c-d)** Lipid profiles and their corresponding transition temperature changes. No significant differences were observed in both cases. Three measurements of each sample were performed. The average is displayed in a thin line; while the shadow band corresponds to the standard deviation. See the text for details.

subunits with the cytoplasm. Then, the activation of cell wall synthesis starts and finishes with the generation of two new daughter cells [34–37]. We associate the existence of the metabolic valley with these cellular mechanisms that are very active mainly in non-incubated cultures [25–27].

To confirm our assertion about the nature of the metabolic valley, we performed some additional experiments to inactivate bacteria. The first consisted of cultures where 100 ml of NaClO (bleach) at 725 mM was added in the incubation process (IB+NaClO). The second sample was a culture incubated with 100 $\mu$L of ampicillin (A) at 715 mM (IB+A). The calorimetry spectra, enthalpy changes, and viability of these samples are shown in S1–S4 Figs. There is a significant

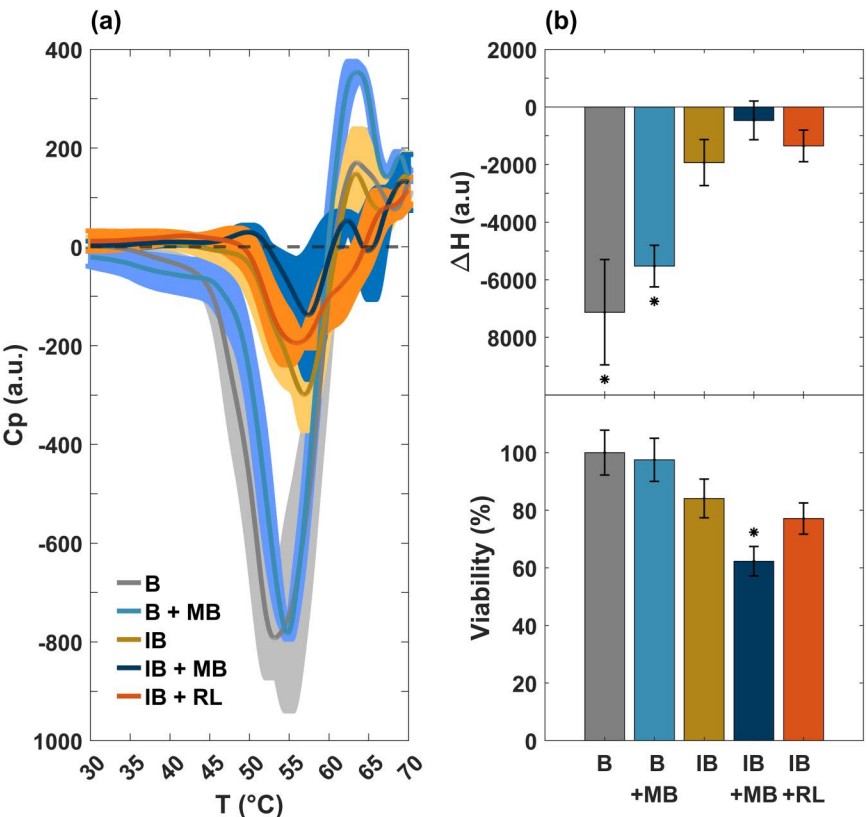

**Fig 7. A detailed approach of the aging time effect for non-incubated (NIB) and incubated (IB) cultures in the zone of the metabolic valley.**
**(a)** Amplification of the valley (downward peak) in the DSC profile. See Fig 5 for details. **(b)** Enthalpy changes and **(c)** viability of non-incubated (NIB) and incubated (IB) samples obtained by differential scanning calorimetry. **(a)** Enthalpy changes of the NIB and NIB+MB cultures show a statistically significant difference respect to the IB control. This finding is consistent with the heat capacity results **(a)**, as these controls exhibit a more pronounced exothermic peak. **(b)** The viability of IB+MB condition is significantly reduced compared to the control (IB). This indicates that photosensitizer alone exhibits a cytotoxic effect after the incubation.

reduction in bacteria treated with NaClO and the disappearance of the metabolic valley in ampicillin-treated cultures, which is related to loss of viability (see S1 and S3 Figs). It is well known that NaClO acts as a disinfectant, and FTsZ has been validated as a relevant target for some antibiotics [34]. It can be seen that only in the first case (IB+NaClO) did the bacteria show significant changes in the protein profile (see S2 Fig) These results can be explained because ampicillin prevents the synthesis of peptidoglycan during bacterial replication but does not denature proteins [38]. However, ampicillin causes rigidity of the bacteria membrane [29–32], as can be observed in S4 Fig, where the calorimetry signal of the lipids is shown. This is a fundamental structure for the proper functioning of the cell. Therefore, it is consistent with the fact that cell viability is affected (see S3 Fig) because the membrane is disturbed and there is an important transition in the temperature of the lipids [29–32] (see S4 Fig).

Now, we can establish a relationship between viability and metabolic valley enthalpy changes corresponding to control and experimental samples (Fig 6 and S3 Fig), which is depicted in Fig 8. This relationship is described by a logarithmic fit using the Eq (1):

$$f = (N_a)/(1 + e^{\alpha\beta(\Delta H_c - \Delta H)})^{(1/\beta)},$$

(1)

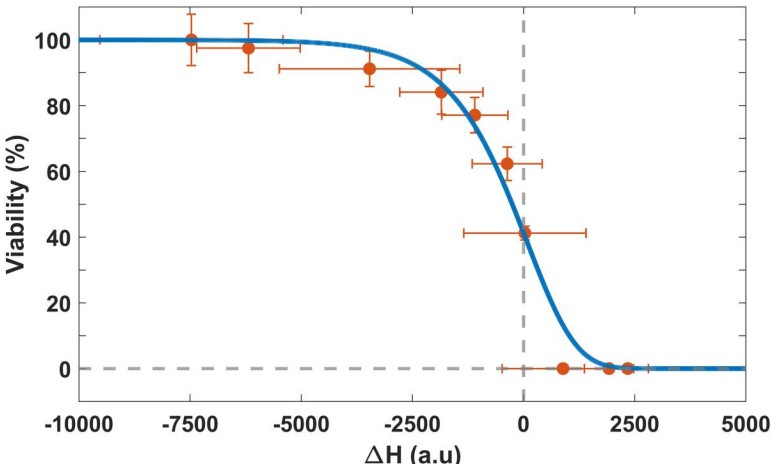

**Fig 8. Logarithmic fit (Richard's model) relating the enthalpy values of the exothermic valleys with the viability of control and photoinacti-vated bacteria.** The r value of the fit is 0.98548. The points represent the average of at least three independent measurements (see Fig 6b and S3 Fig) The error bars are the standard deviation, and the thin line represents the logarithmic fit to the data.

where f is the percentage of viability of the bacteria, $N_a$ = 100 is the upper asymptote (%), $\beta$ = 71.5x10$^{-3}$ is an asymmetry value (dimensionless), $\alpha$ = -17x10$^{-3}$ is the increase rate (1/J), and $\Delta H_c$ = 3x10$^3$ is the enthalpy value (J) when viability is zero.

## Conclusion

In this work, we performed a thermal structure analysis of *E.coli* cultures subjected to the PDI method to improve our understanding of the process that leads to cell death. Highlighting the potential of DSC as a robust tool for understanding biomolecular changes in microorganisms in response to PDI, we found that PDI caused irreversible damage to lipids and proteins, exhibited by stiffening of the lipid membranes and disturbances in protein unfolding regions. We also found an exothermic valley in the control samples, which disappeared after PDI treatment. Finally, we confirm that UVA radiation does have an impact on endogenous photosensitizer molecules, which ultimately led to the death of bacteria.

## Supporting information

**S1 Fig. Calorimetry profiles for alternative bacterial treatments.** Bacteria with sodium hypochlorite (B+NaClO) and bacteria with ampicillin (B+A). There is a significant reduction in the metabolic valley in bacteria treated with NaClO and its disappearance in ampicillin cultures. Three measurements of each sample were performed. The average is displayed in a thin line, while the shadow band corresponds to the standard deviation.
(JPG)

**S2 Fig. Protein profiles and enthalpies in *E.coli* cultures under alternative treatments.** (a) Calorimetry profiles of the protein region for bacteria treated with sodium hypochlorite (B+NaClO) and bacteria with antibiotic (B+A) compared to control (IB). (b) Enthalphy change differences with respect to the control sample showed a reduction in significance only for the first case. Three measurements of each sample were performed. The average is displayed in a thin line, while the shadow band corresponds to the standard deviation.
(JPG)

**S3 Fig. Correlation between viability and cellular energetics of *E.coli* cultures under alternative treatments.** (a) Amplification of the exothermic valley. There is a significant reduction in the metabolic valley in bacteria treated with NaClO (B + NaClO) and its disappearance in cultures with ampicillin (B + A). (b) Comparison between the enthalpy change differences and viability. The treatments exhibit a similar behavior. Three measurements of each sample were performed. (JPG)

**S4 Fig. Lipid profiles and their transition temperature of *E.coli* cultures under alternative treatments.** Calorimetry profiles of the region of the lipid membrane for bacteria treated with sodium hypochlorite (B + NaClO) and bacteria with antibiotic (B + A) compared to control (IB). The average is displayed in a thin line; while the shadow band corresponds to the standard deviation. $\Delta T_m$ corresponds to a change in the melting temperature of the lipid of the bacteria. Positive values mean a stiffening of the lipid membrane. The experiments were performed by triplicate. *$p \leq 0.1$ by the Shapiro-Wilk test and compared to IB.
(JPG)

## Acknowledgments

The authors thank C. Ruiz for providing nanoDSC facilities and M. Márquez-López for some optical measurements. DOZ and CLY received a SECIHTI scholarship.

## Author contributions

**Conceptualization:** Daniel Ortega-Zambrano, Francisco J. Sierra-Valdez, Hilda Mercado-Uribe.

**Data curation:** Daniel Ortega-Zambrano.

**Formal analysis:** Daniel Ortega-Zambrano, Francisco J. Sierra-Valdez, Hilda Mercado-Uribe.

**Funding acquisition:** Hilda Mercado-Uribe.

**Investigation:** Hilda Mercado-Uribe.

**Methodology:** Daniel Ortega-Zambrano, Citlalli Lona-Yepez.

**Project administration:** Hilda Mercado-Uribe.

**Resources:** Hilda Mercado-Uribe.

**Software:** Daniel Ortega-Zambrano, Citlalli Lona-Yepez.

**Supervision:** Francisco J. Sierra-Valdez, Hilda Mercado-Uribe.

**Validation:** Daniel Ortega-Zambrano, Francisco J. Sierra-Valdez.

**Visualization:** Francisco J. Sierra-Valdez, Hilda Mercado-Uribe.

**Writing – original draft:** Hilda Mercado-Uribe.

**Writing – review & editing:** Francisco J. Sierra-Valdez, Hilda Mercado-Uribe.

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
