## [Decision Letter · Decision Letter 0]

6 Jan 2026

Dear Dr. Mercado-Uribe,

Thank you for submitting your manuscript to PLOS ONE. After careful consideration, we feel that it has merit but does not fully meet PLOS ONE’s publication criteria as it currently stands. Therefore, we invite you to submit a revised version of the manuscript that addresses the points raised during the review process.

We look forward to receiving your revised manuscript.

Kind regards,

Amitava Mukherjee, ME, Ph.D.

Academic Editor

PLOS One

Journal Requirements:

https://journals.plos.org/plosone/s/file?id=ba62/PLOSOne_formatting_sample_title_authors_affiliations.pdf....

“This work was funded by

CONACyT, Mexico, Grant number A1-S-8125 (HMU). DOZ and CLY were supported by

fellowships by CONACyT, Mexico.

https://secihti.mx/

The funders had

no role in study design, data collection and analysis, decision to publish, or preparation of the manuscript.”

“This work was supported by CONACyT, Mexico, under Grant A1-S-8125. We acknowledge C. Ruiz for providing nanoDSC facilities. DOZ and CLY received a CONACyT scholarship.”

“This work was funded by CONACyT, Mexico, Grant number A1-S-8125 (HMU). DOZ and CLY were supported byfellowships by CONACyT, Mexico.

https://secihti.mx/

Reviewers' comments:

Reviewer's Responses to Questions

**Comments to the Author**

1. Is the manuscript technically sound, and do the data support the conclusions?

Reviewer #1: Partly

2. Has the statistical analysis been performed appropriately and rigorously?

Reviewer #1: No

3. Have the authors made all data underlying the findings in their manuscript fully available?

Reviewer #1: Yes

4. Is the manuscript presented in an intelligible fashion and written in standard English?

Reviewer #1: No

Reviewer #1: The manuscript explores the thermal behavior of E. coli subjected to photodynamic inactivation (PDI) using differential scanning calorimetry. This is an interesting topic, and whole-cell DSC remains an underutilized yet powerful approach for probing structural and biophysical changes in bacteria. The manuscript contains several promising observations, particularly the correlation between calorimetric features and cell viability, and the exploration of endogenous photosensitizers activated by light.

However, several critical issues limit the robustness of the conclusions in its current form.

1. While the experimental concept is solid, several aspects of the methodology require strengthening.

The photodynamic treatments are not fully characterized: irradiation parameters, light dose, spectral output, and temperature control during 24-hour illumination are insufficiently documented. Given the long irradiation times, even mild light sources can induce thermal artifacts, which may influence DSC outcomes. The interpretation of the “metabolic valley” as a direct marker of metabolic activity is speculative and should be presented more cautiously. Exothermic events during DSC likely reflect irreversible thermal transitions rather than active metabolism at physiological temperatures. Additionally, some mechanistic claims, particularly those related to lipid rigidification, protein destabilization, and endogenous chromophores, would benefit from more quantitative analysis and comparison to existing calorimetric literature.

2. Although nonparametric tests were used, the statistical framework needs improvement. The significance threshold of p ≤ 0.1 is unconventional and should be justified or revised. Error bars in several figures overlap extensively, suggesting weaker statistical power than claimed. A more rigorous statistical presentation, effect sizes, clearer reporting of replicates, and justification for thresholds, would greatly strengthen the manuscript.

3. The manuscript is generally understandable but requires moderate revision to improve clarity, precision, and flow. Some sections (e.g., the discussion of FtsZ/Z-ring dynamics) are overly detailed relative to their relevance for DSC. Several grammatical and stylistic issues should be corrected during revision to ensure unambiguous interpretation.

In summary, the study has merit and contributes to the field of biophysics and photodynamic inactivation. However, to be suitable for publication, the manuscript requires a substantial revision focusing on: i) more detailed characterization of irradiation conditions and controls, ii) more cautious interpretation of DSC features, iii) improved statistical rigor, and iv) clearer, more concise presentation of mechanisms and biological context.

Addressing these issues should significantly enhance the scientific robustness and clarity of the work.

.

Reviewer #1: No

---

## [Author Response · Author response to Decision Letter 1]

11 Mar 2026

February 19th, 2026

Dear Editor,

Please find enclosed a revised version of our manuscript: “Biomolecules involved in the metabolism of Escherichia coli affected by photodynamics: a calorimetry study.”, by D. Ortega-Zambrano, C. Lona-Yepez, F.J. Sierra-Valdez and H. Mercado-Uribe, for your consideration in PlosOne as a research article.

We would like to thank the reviewer for his/her time to carefully revise our article. The manuscript has been modified considering the comments and suggestions made. We also have corrected the errors and improved the paper according to the observed points.

See below our answers and a list of changes.

We hope that with these changes you may judge that our manuscript is now ready to be published in your journal.

Sincerely,

Hilda Mercado-Uribe

Reviewer #1: The manuscript explores the thermal behavior of E. coli subjected to photodynamic inactivation (PDI) using differential scanning calorimetry. This is an interesting topic, and whole-cell DSC remains an underutilized yet powerful approach for probing structural and biophysical changes in bacteria. The manuscript contains several promising observations, particularly the correlation between calorimetric features and cell viability, and the exploration of endogenous photosensitizers activated by light. However, several critical issues limit the robustness of the conclusions in its current form.

Authors:

We appreciate the careful revision of the referee, and the comment that our manuscript addresses an interesting topic. Below, we answer each one of the questions and comments:

1. While the experimental concept is solid, several aspects of the methodology require strengthening. The photodynamic treatments are not fully characterized: irradiation parameters, light dose, spectral output, and temperature control during 24-hour illumination are insufficiently documented.

Authors:

Thank you very much for your observation. We have added more details about the photodynamic treatment (new section).

Given the long irradiation times, even mild light sources can induce thermal artifacts, which may influence DSC outcomes.

Authors:

We agree. It is important to consider that irradiation experiments are long, and the heating of the sample is very likely. For this reason, we first monitored the temperature by a thermocouple (NI USB-TC01) during the irradiation. The temperature increased approximately 4.4 °C at the end of the process. Hence, the thermomagnetic stirrer setpoint was adjusted to 32.6 °C (4.4 °C below the culture temperature) to compensate for the irradiation-induced thermal rise. Additionally, as we have mentioned in the manuscript, to minimize the effect of evaporation, the wells of the empty spaces in the microplate were filled with 1 mL milli-Q water. The microplate was then covered and sealed with parafilm. At the end of the experiment, the final volume of the wells with sample was measured, and the evaporated volume was replenished by adding milli-Q water to restore the initial volume (1 mL). In the current version we include more details about this point (see the Control and experimental section).

The interpretation of the “metabolic valley” as a direct marker of metabolic activity is speculative and should be presented more cautiously. Exothermic events during DSC likely reflect irreversible thermal transitions rather than active metabolism at physiological temperatures.

Authors:

It is well-known that all the reactions that occur within cells during metabolism produce heat and is possible to investigate changes in their metabolic activity through the cell temperature, see Biological thermodynamics, D.T. Haynie, Cambridge, N. Lago. et al. J Therm Anal Calorim., 2011, Microbial electrochemical technology, Chapter 2, Elservier, 2028. In order to do that, it is common to use isothermal microcalorimetry, in which the heat released or absorbed for the sample in a chemical reaction is carefully measured (O. Braissant et al., Methods, (76) 2025). DSC is a complementary method to measure the heat flux inside and outside of the sample (cell) in a temperature-controlled environment. Energy absorption processes (endothermic) are exhibited by peaks in the calorimetry profile, for example, when proteins unfold. Meanwhile, energy releasing processes (exothermic) are exhibited by valleys due to dissipation of heat that is generated by intracellular dynamics, for example, cell replication and protein formation. In this work and for the first time, DSC allowed us to study the effects induced by photodynamics in a whole cell, and secondarily, the changes in the metabolic activity. We confirmed this because there is a significant reduction in the valley (which we called metabolic), in bacteria treated with NaClO (B+NaClO) and its disappearance in cultures with ampicillin (B+A) (See fig. S3 in Supporting information). We have extended our discussion of the results (2nd paragraph in the current version) and we have also added the references we mentioned above, which sustain our arguments.

Additionally, some mechanistic claims, particularly those related to lipid rigidification, protein destabilization, and endogenous chromophores, would benefit from more quantitative analysis and comparison to existing calorimetric literature.

Authors:

We thank the referee for this suggestion. We have extensively reviewed the literature and modified the manuscript accordingly, including more details and references concerning these topics. See the section “Results and Discussion”.

2. Although nonparametric tests were used, the statistical framework needs improvement. The significance threshold of p ≤ 0.1 is unconventional and should be justified or revised. Error bars in several figures overlap extensively, suggesting weaker statistical power than claimed. A more rigorous statistical presentation, effect sizes, clearer reporting of replicates, and justification for thresholds, would greatly strengthen the manuscript.

Authors:

We agree that a significant threshold of p ≤ 0.1 is not the standard for confirmatory studies. However, in this work it was intentionally used as an exploratory criterion (90% confidence level) for the following reasons: several conditions were evaluated, five control groups (and two in Supporting information) and two experimental groups. All were analysed in triplicate. Because we had n=3, the Shapiro–Wilk normality test indicated that the requirements for parametric analysis were not met. Then, non-parametric methods were applied, which tend to be more conservatives under this experimental scheme.

It is worth mentioning that there is an inherent variability due to biological nature of the samples. Variability is due to all the different treatments the samples suffer before entering the DSC. However, despite this intrinsic variability, we observe significant differences in the case we are truly interested, that is, in the irradiated samples. Additionally, it is important to note that the scale in Fig.6 was greatly amplified.

3. The manuscript is generally understandable but requires moderate revision to improve clarity, precision, and flow. Some sections (e.g., the discussion of FtsZ/Z-ring dynamics) are overly detailed relative to their relevance for DSC. Several grammatical and stylistic issues should be corrected during revision to ensure unambiguous interpretation.

Authors:

We appreciate your encouraging comments. We have considered your observations, and we have made changes to synthetize the mentioned section about the FtsZ/Z-ring dynamics and carefully edited and improved the current version.

In summary, the study has merit and contributes to the field of biophysics and photodynamic

inactivation. However, to be suitable for publication, the manuscript requires a substantial revision focusing on: i) more detailed characterization of irradiation conditions and controls, ii) more cautious interpretation of DSC features, iii) improved statistical rigor, and iv) clearer, more concise presentation of mechanisms and biological context. Addressing these issues should significantly enhance the scientific robustness and clarity of the work.

Authors: We thank the recommendations of the reviewer.

List of changes:

1. We corrected the abbreviation and link of the funder (the institution changed recently its name). We updated our Funding Statement.

2. Funding information does not appear in the Acknowledgments section now. We removed any funding-related text from the manuscript.

3. We have added more details in the first paragraph of the section “Control and experimental samples”.

4. We have added a new section about the Photodynamic treatment.

5. We have modified some paragraphs in the Results and Discussion section.

6. Several modifications have been made to the text to improve our article.

7. We have added references 17, 18, and 24-32.

8. All figures were checked and inconsistences fixed.

---

## [Decision Letter · Decision Letter 1]

1 Apr 2026

Biomolecules involved in the metabolism of Escherichia coli affected by photodynamics: a calorimetry study.

PONE-D-25-58383R1

Dear Dr. Mercado-Uribe,

We’re pleased to inform you that your manuscript has been judged scientifically suitable for publication and will be formally accepted for publication once it meets all outstanding technical requirements.

Kind regards,

Amitava Mukherjee, ME, Ph.D.

Academic Editor

PLOS One

Additional Editor Comments (optional):

Reviewers' comments:

Reviewer's Responses to Questions

**Comments to the Author**

Reviewer #1: All comments have been addressed

2. Is the manuscript technically sound, and do the data support the conclusions?

Reviewer #1: Yes

3. Has the statistical analysis been performed appropriately and rigorously?

Reviewer #1: Yes

4. Have the authors made all data underlying the findings in their manuscript fully available?

Reviewer #1: Yes

5. Is the manuscript presented in an intelligible fashion and written in standard English?

Reviewer #1: Yes

Reviewer #1: The revised version of the manuscript shows a clear and constructive response to my previous comments. The authors have addressed the major concerns raised in the previous round, and the manuscript has improved both in clarity and scientific rigor. In particular, the revisions better justify the experimental design and provide a more balanced discussion of the results. The previously noted ambiguities have been clarified, and the overall narrative is now more coherent and focused. The added explanations strengthen the interpretation of the data and improve the readability of the manuscript. Only minor editorial adjustments could still be considered (mainly language polishing and slight tightening of some descriptive sections), but these do not affect the scientific quality of the work. The manuscript is now suitable for publication in its current form.

.

Reviewer #1: No

---

## [Editor Report · Acceptance letter]

PONE-D-25-58383R1

PLOS One

Dear Dr. Mercado-Uribe,

I'm pleased to inform you that your manuscript has been deemed suitable for publication in PLOS One. Congratulations! Your manuscript is now being handed over to our production team.

Kind regards,

on behalf of

Professor Dr. Amitava Mukherjee

Academic Editor

PLOS One